# AfriSenti: A Twitter Sentiment Analysis Benchmark for African Languages

**Shamsuddeen Hassan Muhammad**[1,2*+], **Idris Abdulmumin**[3+], **Abinew Ali Ayele**[4],
**Nedjma Ousidhoum**[5,6], **David Ifeoluwa Adelani**[7*], **Seid Muhie Yimam**[8],
**Ibrahim Sa'id Ahmad**[2+], **Meriem Beloucif** [9], **Saif M. Mohammad**[10],
**Sebastian Ruder**[11], **Oumaima Hourrane**[12], **Pavel Brazdil**[13], **Alípio Jorge** [1,13],
**Felermino Dário Mário António Ali**[1], **Davis David**[14], **Salomey Osei**[15], **Bello Shehu Bello**[2],
**Falalu Ibrahim**[16], **Tajuddeen Gwadabe**[*+], **Samuel Rutunda**[17], **Tadesse Belay**[18],
**Wendimu Baye Messelle**[4], **Hailu Beshada Balcha**[19], **Sisay Adugna Chala**[20],
**Hagos Tesfahun Gebremichael**[4], **Bernard Opoku**[21], **Steven Arthur**[21]

[1]University of Porto, Portugal [2]Bayero University Kano, [3]Ahmadu Bello University, Zaria, [4]Bahir Dar University,

[5]University of Cambridge, [6]Cardiff University, [7]University College London, [8]Universität Hamburg, [9]Uppsala University,

[10]National Research Council Canada, [11]Google Research, [12]Hassan II University of Casablanca, [13]LIAAD - INESC TEC,

[14]dLab, [15]University of Deusto, [16]Kaduna State University, [17]Digital Umuganda, [18]Wollo University, [19]Jimma University,

[20]Fraunhofer FIT, [21]Accra Institute of Technology, [*]Masakhane NLP, [+]HausaNLP

shmuhammad.csc@buk.edu.ng

## Abstract

Africa is home to over 2,000 languages from more than six language families and has the highest linguistic diversity among all continents. These include 75 languages with at least one million speakers each. Yet, there is little NLP research conducted on African languages. Crucial to enabling such research is the availability of high-quality annotated datasets. In this paper, we introduce AfriSenti, a sentiment analysis benchmark that contains a total of >110,000 tweets in 14 African languages (Amharic, Algerian Arabic, Hausa, Igbo, Kinyarwanda, Moroccan Arabic, Mozambican Portuguese, Nigerian Pidgin, Oromo, Swahili, Tigrinya, Twi, Xitsonga, and Yorùbá) from four language families. The tweets were annotated by native speakers and used in the AfriSenti-SemEval shared task [1].

We describe the data collection methodology, annotation process, and the challenges we dealt with when curating each dataset. We further report baseline experiments conducted on the different datasets and discuss their usefulness.

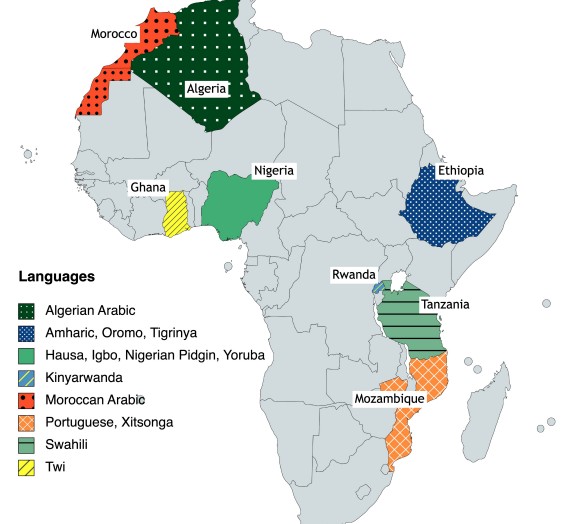

Figure 1: Countries and languages represented in AfriSenti: Amharic, Algerian Arabic, Hausa, Igbo, Kinyarwanda, Moroccan Arabic, Mozambican Portuguese, Nigerian Pidgin, Oromo, Swahili, Tigrinya, Twi, Xitsonga, and Yorùbá.

## 1 Introduction

Africa has a long and rich linguistic history, experiencing language contact, language expansion, development of trade languages, language shift, and language death on several occasions. The continent is incredibly diverse linguistically and is home to over 2,000 languages. These include 75 languages with at least one million speakers each. Africa has a rich tradition of storytelling, poems, songs, and literature (Banks-Wallace, 2002; Carter-Black,

2007) and in recent years, it has seen a proliferation of communication in digital and social media. Code-switching is common in these new forms of communication where speakers alternate between two or more languages in the context of a single conversation (Angel et al., 2020; Thara and Poornachandran, 2018; Santy et al., 2021). However, despite this linguistic richness, African languages have been comparatively under-represented in natural language processing (NLP) research.

An influential sub-area of NLP deals with sentiment, valence, emotions, and affect in language (Liu, 2020). Computational analysis of emotion

---

[1]The AfriSenti Shared Task had over 200 participants. See website: https://afrisenti-semeval.github.io

| Lang. | Tweet | Sentiment |
|-------|-------|-----------|
| amh | ያ ጨካኝ አረመኔ ታስሮ ይሽዉ ካቴና ገብቶላታል ይሱናል። ቆይ አስረዉ የጀበና ቡና አየጋበዙት ነዉ እንዴ? | negative |
| arq | @user .... الشروق هذه من خرجت وهي نتاع تهديل، مستوى منحط وشعبوي | negative |
| ary | واش بغيتوهم يبداو يتكرفسو على العادي والبادي عاد تبقاو أنتا على خاطر خاطركم | negative |
| ary | rabi ykhali alhbiba makayn ghir nachat o chi machat | positive |
| hau | @USER Aunt rahma i luv u wallah irin totally dinnan | positive |
| ibo | akowaro ya ofuma nne kai daalu nwanne mmadu | positive |
| kin | @user Ariko akokanu ngo inyebebe unyujijemo sisawa wangu | negative |
| orm | @user Jawaar Kenya OMN haala akkamiin argachuu dandeenya | neutral |
| por | Honestidade é algo que não se compra. Infelizmente a humanidade esqueceu disso por causa das suas ambições. | positive |
| pcm | E don tay wey I don dey crush on this fine woman … | positive |
| swa | Asante sana watu wa Sirari jimbo la Tarime vijijini Huu ni Upendo usio na Mashaka kwa Mbunge wenu John Heche | positive |
| tir | @user ከመኸረኩም እንተኾይነ:ንሕውሓት ነዚም ውሓ,ድ ቁጽርም እስ ምጥፋእ ይሕሸ ኩም! | negative |
| tso | @user @user Yu , tindzava ? Tsika mbangui mpfana e nita ku despro-gramara | negative |
| twi | messi saf den check en bp na wo kwame danso wo di twe da kor aaa na wawu | negative |
| yor | onírèégbè aláàdúgbò ati olójúkòkòrò | negative |

Table 1: Examples of tweets and their sentiments in the different AfriSenti Languages. Note that the collected tweets in Moroccan Arabic/Darija (ary) are written in both Arabic and Latin scripts. The translations can be found in the Appendix (Table 10).

states in language and the creation of systems that predict these states from utterances have applications in literary analysis and culturomics (Hamilton et al., 2016; Reagan et al., 2016), e-commerce (e.g., tracking feelings towards products), and research in psychology and social science (Dodds et al., 2015; Hamilton et al., 2016). Despite the tremendous amount of work in this important space over the last two decades, there is little work on African languages, partially due to a lack of high-quality annotated data.

To enable sentiment analysis research in African languages, we present **AfriSenti**, the largest sentiment analysis benchmark for under-represented African languages—covering 110,000+ tweets annotated as positive, negative or neutral, in 14 languages[2] (Amharic, Algerian Arabic, Hausa, Igbo, Kinyarwanda, Moroccan Arabic, Mozambican Portuguese, Nigerian Pidgin, Oromo, Swahili, Tigrinya, Twi, Xitsonga, and Yorùbá) from four language families (Afro-Asiatic, English Creole, Indo-European and Niger-Congo)[3]. We show the represented countries and languages in Figure 1

and provide examples of annotated tweets in Table 1. The datasets were used in the first Afrocentric SemEval shared task *SemEval 2023 Task 12: Sentiment analysis for African languages (AfriSenti-SemEval)* (Muhammad et al., 2023). We publicly release the data, which provides further opportunities to investigate the difficulty of sentiment analysis for African languages by e.g., building sentiment analysis systems for various African languages, and studying of sentiment and contemporary language use in these languages.

Our contributions are: (1) the creation of the largest Twitter dataset for sentiment analysis in African languages by annotating ten new datasets and curating four existing ones (Muhammad et al., 2022), (2) the discussion of the data collection and annotation process in 14 low-resource African languages, (3) the release of sentiment lexicons for these languages, (4) the presentation of classification baseline results using our datasets.

## 2 Related Work

Research in sentiment analysis developed since the early days of lexicon-based sentiment analysis approaches (Mohammad et al., 2013; Taboada et al., 2011; Turney, 2002) to more advanced ML

[2]For simplicity, we use the term language to refer to language varieties including dialects.

[3]The datasets are publicly available on https://github.com/afrisenti-semeval/afrisent-semeval-2023

| Language | ISO Code | Subregion | Spoken in | Script |
|----------|----------|-----------|-----------|--------|
| Amharic | amh | East Africa | Ethiopia | Ethiopic |
| Algerian Arabic/Darja | arq | North Africa | Algeria | Arabic |
| Hausa | hau | West Africa | Northern Nigeria, Ghana, Cameroon, | Latin |
| Igbo | ibo | West Africa | Southeastern Nigeria | Latin |
| Kinyarwanda | kin | East Africa | Rwanda | Latin |
| Moroccan Arabic/Darija | ary | North Africa | Morocco | Arabic/Latin |
| Mozambican Portuguese | pt-MZ | Southeastern Africa | Mozambique | Latin |
| Nigerian Pidgin | pcm | West Africa | Nigeria, Ghana, Cameroon, | Latin |
| Oromo | orm | East Africa | Ethiopia | Latin |
| Swahili | swa | East Africa | Tanzania, Kenya, Mozambique | Latin |
| Tigrinya | tir | East Africa | Ethiopia | Ethiopic |
| Twi | twi | West Africa | Ghana | Latin |
| Xitsonga | tso | Southern Africa | Mozambique, South Africa, Zimbabwe, Eswatini | Latin |
| Yorùbá | yor | West Africa | Southwestern and Central Nigeria | Latin |

Table 2: African languages included in our study (Lewis, 2009). For each language, we report its ISO code, the African sub-regions it is mainly spoken in, and the writing scripts included in its dataset collection.

approaches (Agarwal and Mittal, 2016; Le and Nguyen, 2020), deep learning-based methods (Yadav and Vishwakarma, 2020; Zhang et al., 2018), and hybrid approaches (Gupta and Joshi, 2020; Kaur et al., 2022). Nowadays, Pretrained Language Models (PLMs), e.g., XLM-R (Conneau et al., 2020), mDeBERTaV3 (He et al., 2021), AfriBERTa (Ogueji et al., 2021b), AfroXLMR (Alabi et al., 2022) and XLM-T (Barbieri et al., 2022b), help us achieve state-of-the-art performance for this task.

Recent work in sentiment analysis focused on subtasks that tackle new challenges, including aspect-based (Chen et al., 2022), multimodal (Liang et al., 2022), explainable (neuro-symbolic) (Cambria et al., 2022), and multilingual sentiment analysis (Muhammad et al., 2022). On the other hand, standard sentiment analysis subtasks such as polarity classification (positive, negative, neutral) are widely considered saturated and solved (Poria et al., 2020), with an accuracy of 97.5% in certain domains (Jiang et al., 2020; Raffel et al., 2020). However, while this may be true for high-resource languages in relatively clean, long-form text domains such as movie reviews, noisy user-generated data in under-represented languages still presents a challenge (Yimam et al., 2020). Additionally, African languages present other difficulties for sentiment analysis such as dealing with tone, code-switching, and digraphia (Adebara and Abdul-Mageed, 2022). Existing work in sentiment analysis for African languages has therefore mainly focused on polarity classification (El Abdouli et al.,

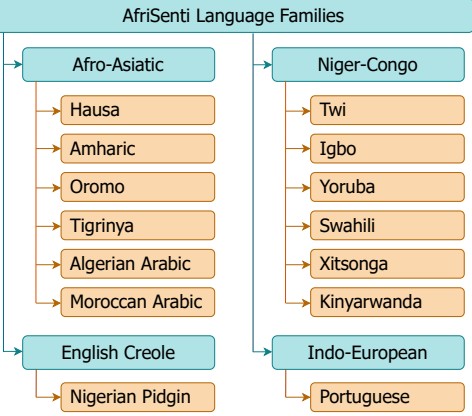

Figure 2: The language Family (in green) of each language (in yellow) included in AfriSenti.

2017; Martin et al., 2021; Mataoui et al., 2016; Moudjari et al., 2020; Muhammad et al., 2022; Yimam et al., 2020). Our benchmark, AfriSenti, is the largest multilingual dataset for sentiment analysis in African languages.

## 3 Overview of the AfriSenti Datasets

AfriSenti covers 14 African languages (see Table 2), each with unique linguistic characteristics and writing systems. As shown in Figure 2, the benchmark includes six languages of the Afro-Asiatic family, six languages of the Niger-Congo family, one from the English Creole family, and one from the Indo-European family.

**Writing Systems** Scripts serve not only as a means of transcribing spoken language, but also as powerful cultural symbols that reflect people's

identity (Sterponi and Lai, 2014). For instance, the Bamun script is deeply connected to the identity of Bamun speakers in Cameroon, while the Geez/Ethiopic script (for Amharic and Tigrinya) evokes the strength and significance of Ethiopian culture (Sterponi and Lai, 2014). Similarly, the Ajami script, a variant of the Arabic script used in various African languages such as Hausa, serves as a reminder of the rich African cultural heritage of the Hausa community (Gee, 2005).

African languages, with a few exceptions, use the Latin script, written from left to right, or the Arabic script, written from right to left (Gee, 2005; Meshesha and Jawahar, 2008), with the Latin script being the most widely used in Africa (Eberhard et al., 2020). Ten languages out of fourteen in AfriSenti are written in Latin script, two in Arabic script, and two in Ethiopic (or Geez) script. On social media, people may write Moroccan Arabic (Moroccan Darija) and Algerian Arabic (Algerian Darja) in both Latin and Arabic characters due to various reasons including access to technology, i.e., the fact that Arabic keyboards were not easily accessible on commonly used devices for many years, code-switching, and other phenomena. This makes Algerian and Moroccan Arabic digraphic, i.e., their texts can be written in multiple scripts on social media [4]. Similarly, Amharic is digraphic and is written in both Latin and Geez script (Belay et al., 2021).

**Geographic Representation** AfriSenti covers the majority of African sub-regions. Many African languages are spoken in neighbouring countries within the same sub-regions. For instance, variations of Hausa are spoken in Nigeria, Ghana, and Cameroon, while Swahili variants are widely spoken in East African countries, including Kenya, Tanzania, and Uganda. AfriSenti also includes datasets in the top three languages with the highest numbers of speakers in Africa (Swahili, Amharic, and Hausa). Figure 1 shows the geographic distribution of the languages represented in AfriSenti.

**New and Existing Datasets** AfriSenti includes existing and newly created datasets as shown in Table 3. For the existing datasets whose test sets are public, we created new test sets to further evaluate their performance in the AfriSenti-SemEval shared

| Lang. | New | Existing | Source |
|---|---|---|---|
| ama | test | train, dev | Yimam et al. (2020) |
| arq | all | ✗ | - |
| ary | all | ✗ | - |
| hau | ✗ | all | Muhammad et al. (2022) |
| ibo | ✗ | all | Muhammad et al. (2022) |
| kin | all | ✗ | - |
| orm | all | ✗ | - |
| pcm | ✗ | all | Muhammad et al. (2022) |
| pt-MZ | all | ✗ | - |
| swa | all | ✗ | - |
| tir | all | ✗ | - |
| tso | all | ✗ | - |
| twi | all | ✗ | - |
| yor | ✗ | all | Muhammad et al. (2022) |

Table 3: The AfriSenti datasets. We show the new and previously available datasets (with their sources).

task.

## 4 Data Collection and Processing

**Twitter's Limited Support for African Languages** Since many people share their opinions on Twitter, the platform is widely used to study sentiment analysis (Muhammad et al., 2022). However, the Twitter API's support for African languages is limited [5], which makes it difficult for researchers to collect data. Specifically, the Twitter language API supports only Amharic out of more than 2,000 African languages[6]. This disparity in language coverage highlights the need for further research and development in NLP for low-resource languages.

### 4.1 Tweet Collection

We used the Twitter Academic API to collect tweets. However, as the API does not provide language identification for tweets in African languages, we used location-based and vocabulary-based heuristics to collect the datasets.

### 4.1.1 Location-based data collection

For all languages except Algerian Arabic and Afaan Oromo, we used a location-based collection approach to filter out results. Hence, tweets were collected based on the names of the countries where the majority of the target language speakers

---

[4]Table 1 shows an example of Moroccan Arabic/Darija tweets written in Latin and Arabic script. For Algerian Arabic/Darja and Amharic, AfriSenti includes data in only Arabic and Geez scripts.

[5]The data collection process was conducted before December $20^{th}$, 2022. I.e., before the change of policy that took place in 2023.

[6]https://blog.twitter.com/engineering/en_us/a/2015/evaluating-language-identification-performance

| Lang. | Manually | Translated | Source |
|---|---|---|---|
| ama | ✓ | ✗ | Yimam et al. (2020) |
| arq | ✓ | ✗ | - |
| hau | ✓ | ✓ | Muhammad et al. (2022) |
| ibo | ✓ | ✓ | Muhammad et al. (2022) |
| ary | ✗ | ✗ | - |
| orm | ✓ | ✗ | Yimam et al. (2020) |
| pcm | ✓ | ✗ | Muhammad et al. (2022) |
| pt-MZ | ✓ | ✗ | - |
| kin | ✗ | ✓ | - |
| swa | ✗ | ✗ | - |
| tir | ✓ | ✗ | Yimam et al. (2020) |
| tso | ✗ | ✓ | - |
| twi | ✗ | ✗ | - |
| yor | ✓ | ✓ | Muhammad et al. (2022) |

Table 4: Manually collected and translated lexicons in AfriSenti.

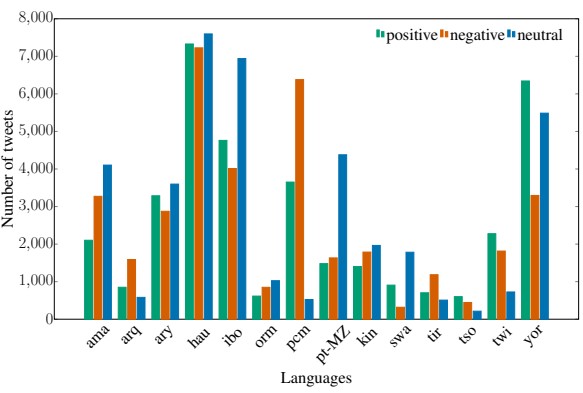

Figure 3: Label distributions for the different AfriSenti datasets (i.e., number of positive, negative, and neutral tweets).

are located. For Afaan Oromo, tweets were collected globally due to the small size of the initial data collected from Ethiopia.

#### 4.1.2 Vocabulary-based Data Collection

As different languages are spoken within the same region in Africa (Amfo and Anderson, 2019), the location-based approach did not help in all cases. For instance, searching for tweets from *"Lagos"* (Nigeria) returned tweets in multiple languages, such as *Yorùbá*, *Igbo*, *Hausa*, *Pidgin*, *English*, etc.

To address this, we combined the location-based approach with vocabulary-based collection strategies. These included the use of stopwords, sentiment lexicons, and a language detection tool. For languages that used the Geez script, we used the Ethiopic Twitter Dataset for Amharic (ETD-AM), which includes tweets that were collected since 2014 (Yimam et al., 2019).

**Data collection using stopwords** Most African languages do not have curated stopword lists (Emezue et al., 2022). Therefore, we created stopword lists for some AfriSenti languages and used them to collect data. We used corpora from different domains, i.e., news data and religious texts, to rank words based on their frequency (Adelani et al., 2021). We filtered out the top 100 words by deleting domain-specific words (e.g., the word *God* in religious texts) and created lists based on the top 50 words that appeared across domains.

We also used a word co-occurrence-based approach to extract stopwords (Liang et al., 2009) using text sources from different domains. We lower-cased and removed punctuation marks and numbers, constructed a co-occurrence graph, and filtered out the words that occurred most often. Na-

tive speakers verified the generated lists before use. This approach worked the best for Xistonga.

**Data collection using sentiment lexicons** As data collection based on stopwords sometimes results in tweets that are inadequate for sentiment analysis (e.g., too many neutral tweets), we used a sentiment lexicon—a dictionary of positive and negative words—for tweet collection. This allows for a balanced collection across sentiment classes (positive/negative/neutral). For Moroccan Arabic, we used emotion word list curated by Outchakoucht and Es-Samaali (2021).

Table 4 provides details on the sentiment lexicons in AfriSenti and indicates whether they were manually created or translated.

**Data collection using mixed lists of words** Besides stopwords and sentiment lexicons, native speakers provided lists of language-specific terms including generic words. For instance, this strategy helped us collect Algerian Arabic tweets, and the generic terms included equivalents of words such as " الغاشي " *(the crowd)* and names of Algerian cities.

Figure 3 shows the distribution of the three sentiment labels (i.e., positive, negative, and neutral) for each language.

### 4.2 Language Detection

As we mainly used heuristics for data collection, the collected tweets included some in different languages. For instance, when collecting tweets using lists of Amharic words, some returned tweets were in Tigrinya, due to Amharic–Tigrinya code-mixing. Similarly, when searching for Algerian Arabic tweets, Tunisian, Moroccan, and Modern Standard Arabic tweets were found due to overlap-

ping terms. Hence, we used different techniques for language detection as a post-processing step.

**Language detection using existing tools**   Few African languages have pre-existing language detection tools (Keet, 2021). We used Google CLD3[7] and the Pycld2 library[8] for the supported AfriSenti languages (Amharic, Oromo and Tigrinya).

**Manual language detection**   For languages that do not have a pre-existing tool, the detection was conducted by native speakers. For instance, annotators who are native speakers of Twi and Xitsonga manually labeled 2,000 tweets in these languages. In addition, as native speakers collected the Algerian Arabic tweets, they deleted all possible tweets expressed in another language or a different Arabic variation.

**Language detection using pre-trained language models**   To reduce the effort spent on language detection, we also used a pre-trained language model fine-tuned on 2,000 manually annotated tweets (Caswell et al., 2020) to identify Twi and Xitsonga. Despite our efforts to detect the languages, we note that as multilingualism is common in African societies, the final dataset contains some code-mixed tweets.

### 4.3   Tweet Anonymization and Pre-processing

We anonymized the tweets by replacing all @*mentions* by @*user* and removed all URLs. For the Nigerian language test sets, we further lower-cased the tweets (Muhammad et al., 2022).

## 5   Data Annotation Challenges

Tweet samples were randomly selected based on the different collection strategies. Then, with the exception of the Ethiopian languages, each tweet was annotated by three native speakers.

We used the *Simple Sentiment Questionnaire* annotation guide by Mohammad (2016) and used majority voting (Davani et al., 2021) to determine the final sentiment label for each tweet (Muhammad et al., 2022). We discarded the cases where all annotators disagreed. The datasets of the three Ethiopian languages (Amharic, Tigriniya, and Oromo) were annotated using two independent annotators, and then curated by a third more experienced individual who decided on the final gold labels.

Prabhakaran et al. (2021) showed that the majority vote conceals systematic disagreements between annotators resulting from their sociocultural backgrounds and experiences. Therefore, we release all the individual labels to the research community. We report the free marginal multi-rater kappa scores (Randolph, 2005) in Table 5 since chance-adjusted scores such as Fleiss-$\kappa$ can be low despite a high agreement due to the imbalanced label distributions (Falotico and Quatto, 2015; Matheson, 2019; Randolph, 2005). We obtained intermediate to good levels of agreement ($0.40 - 0.75$) across all languages, except for Afaan Oromo where we obtained a relatively low agreement score due to the annotation challenges that we discuss in Section 5.

Table 6 shows the number of tweets in each of the 14 datasets. The Hausa collection of tweets is the largest among all the datasets and the Xitsonga dataset is the smallest one. Figure 3 shows the distribution of the labeled classes in the datasets. We observe that the distribution for some languages such as ha is fairly equitable while in others such as pcm, the proportion of tweets in each class varies widely. Sentiment annotation for African languages presents some challenges (Muhammad et al., 2022) that we highlight in the following.

**Hausa, Igbo, and Yorùbá**   Hausa, Igbo, and Yorùbá are tonal languages, which contributes to the difficulty of the task as the tone is rarely fully rendered in written form. E.g., in Hausa, *Bàaba* means dad and *Baabà* means mum (typically written *Baba* on social media), and in Yorùbá, *èdè* means language, and *edé* means crayfish (typically written *ede* on social media).

Further, the intonation, which is crucial to the understanding of the tweet may not be conveyed in the text. E.g., *ò nwèkwàrà mgbe i naenwe sense ? (will you ever be able to talk sensibly? – You're a fool.)* and *ò nwèkwàrà mgbe i naenwe sense ( sometimes you act with great maturity. – I'm impressed.)* are almost identical but carry different sentiments (i.e., negative and positive, respectively). In this case, the difference in the intonation may not be clear either due to the (non)use of punctuation or the lack of context.

**Twi**   A significant portion of tweets in *Twi* were ambiguous, making it difficult to categorize sentiment accurately. Some tweets contained symbols not in the Twi alphabet, which is a frequent oc-

---

[7]https://github.com/google/cld3
[8]https://pypi.org/project/pycld2/

| | 3-way | | | | | | | | | | | 2-way | | |
|---|---|---|---|---|---|---|---|---|---|---|---|---|---|---|
| **Lang.** | **arq** | **ary** | **hau** | **ibo** | **kin** | **pcm** | **pt-MZ** | **swa** | **tso** | **twi** | **yor** | **ama** | **orm** | **tir** |
| $\kappa$ | 0.41 | 0.62 | 0.66 | 0.61 | 0.43 | 0.60 | 0.50 | - | 0.50 | 0.51 | 0.65 | 0.47 | 0.20 | 0.51 |

Table 5: Inter-annotator agreement scores using the free marginal multi-rater kappa (Randolph, 2005) for the different languages.

| | **ama** | **arq** | **hau** | **ibo** | **ary** | **orm** | **pcm** | **pt-MZ** | **kin** | **swa** | **tir** | **tso** | **twi** | **yor** |
|---|---|---|---|---|---|---|---|---|---|---|---|---|---|---|
| **train** | 5,985 | 1,652 | 14,173 | 10,193 | 5,584 | - | 5,122 | 3,064 | 3,303 | 1,811 | - | 805 | 3,482 | 8,523 |
| **dev** | 1,498 | 415 | 2,678 | 1,842 | 1,216 | 397 | 1,282 | 768 | 828 | 454 | 399 | 204 | 389 | 2,091 |
| **test** | 2,000 | 959 | 5,304 | 3,683 | 2,962 | 2,097 | 4,155 | 3,663 | 1,027 | 749 | 2,001 | 255 | 950 | 4,516 |
| **Total** | 9,483 | 3,062 | 22,155 | 15,718 | 9,762 | 2,494 | 10,559 | 7,495 | 5,158 | 3,014 | 2,400 | 1,264 | 4,821 | 15,130 |

Table 6: Splits and sizes of the AfriSenti datasets. We do not allocate training splits for Afaan Oromo (orm) and Tigrinya (tir) due to the limited size of the data and only evaluate on them in a zero-shot transfer setting in §6.

currence due to the lack of support for certain Twi letters on keyboards (Scannell, 2011). For example, "ɔ" is replaced by the English letter "c", and "ɛ" is replaced by "3".

Additionally, tweets were often annotated as negative (cf. Figure 3) due to some common expressions that could be seen as offensive depending on the context. E.g., "*Tweaa*" was once considered an insult but has become a playful expression through trolling, and "*gyae gyimii*" is commonly used by young people to say "stop" while its literal meaning is "stop fooling".

**Mozambican Portuguese and Xitsonga**  One of the significant challenges for the Mozambican Portuguese and Xitsonga data annotators was the presence of code-mixed and sarcastic tweets. Code-mixing in tweets made it challenging for the annotators to determine the intended meaning of the tweet as it involved multiple languages spoken in Mozambique that some annotators were unfamiliar with. Similarly, the presence of two variants of Xitsonga spoken in Mozambique (Changana and Ronga) added to the complexity of the annotation task. Additionally. we excluded many tweets from the final dataset as sarcasm present in tweets was another source of disagreement among the annotators.

**Ethiopian languages**  For Afaan Oromo and Tigrinya, challenges included finding annotators and the lack of a reliable Internet connection and access to personal computers. Although we trained the Oromo annotators, we observed severe problems in the quality of the annotated data, which led to a low agreement score.

**Algerian Arabic**  For Algerian Arabic, the main challenge was the use of sarcasm. When this caused a disagreement among the annotators, the tweet was further labeled by two other annotators. If the annotators did not agree on one final label, the tweet was discarded. As Twitter is also commonly used to discuss controversial topics in the region, we found a large number of offensive tweets shared among the users. We removed the offensive tweets to protect the annotators and avoid including such instances in a sentiment analysis dataset.

## 6 Experiments

### 6.1 Setup

For our baseline experiments, we considered three settings: (1) monolingual baseline models based on multilingual pre-trained language models for 12 AfriSenti languages with training data, (2) multilingual training of all 12 languages and their evaluation on a combined test of all 12 languages, (3) zero-shot transfer to Oromo (orm) and Tigrinya (tir) from any of the 12 languages with available training data. We used a standard configuration for text classification fine-tuning on HuggingFace with a learning rate of $2e-5$ for smaller PLMs and $1e-5$ for larger PLMs, a batch size of 128, and 10 epochs.

**Monolingual baseline models**  We fine-tune massively multilingual PLMs trained on 100 languages from around the world as well as Africa-centric PLMs trained exclusively on languages spoken in Africa. For the massively multilingual PLMs, we selected two representative PLMs: XLM-R-{base & large} (Conneau et al., 2020) and mDeBER-

| Lang. | In XLM-R or mDeBERTa? | In AfriBERTa | In AfroXLMR | In XLM-T | AfriBERTa large | XLM-R base | AfroXLMR base | mDeBERTa base | XLM-T base | XLM-R large | AfroXLMR large |
|---|---|---|---|---|---|---|---|---|---|---|---|
| amh | ✓ | ✓ | ✓ | ✓ | 56.9 | 60.2 | 54.9 | 57.6 | 60.8 | **61.8** | 61.6 |
| arq | ✓ | ✗ | ✓ | ✓ | 47.7 | 65.9 | 65.5 | 65.7 | **69.5** | 63.9 | 68.3 |
| ary | ✓ | ✗ | ✓ | ✓ | 44.1 | 50.9 | 52.4 | 55.0 | **58.3** | 57.7 | 56.6 |
| hau | ✓ | ✓ | ✓ | ✗ | 78.7 | 73.2 | 77.2 | 75.7 | 73.3 | 75.7 | **80.7** |
| ibo | ✗ | ✓ | ✓ | ✗ | 78.6 | 75.6 | 76.3 | 77.5 | 76.1 | 76.5 | **79.5** |
| kin | ✗ | ✓ | ✓ | ✗ | 62.7 | 56.7 | 67.2 | 65.5 | 59.0 | 55.7 | **70.6** |
| pcm | ✗ | ✓ | ✓ | ✗ | 62.3 | 63.8 | 67.6 | 66.2 | 66.6 | 67.2 | **68.7** |
| pt-MZ | ✓ | ✗ | ✗ | ✓ | 58.3 | 70.1 | 66.6 | 68.6 | 71.3 | 71.6 | **71.6** |
| swa | ✓ | ✓ | ✓ | ✗ | 61.5 | 57.8 | 60.8 | 59.5 | 58.4 | 61.4 | **63.4** |
| tso | ✗ | ✗ | ✗ | ✗ | 51.6 | 47.4 | 45.9 | 47.4 | **53.8** | 43.7 | 47.3 |
| twi | ✗ | ✗ | ✗ | ✗ | **65.2** | 61.4 | 62.6 | 63.8 | 65.1 | 59.9 | 64.3 |
| yor | ✗ | ✓ | ✓ | ✗ | 72.9 | 62.7 | 70.0 | 68.4 | 64.2 | 62.4 | **74.1** |
| AVG | - | - | - | - | 61.7 | 61.9 | 63.9 | 64.2 | 64.7 | 63.1 | **67.2** |

Table 7: Accuracy scores of monolingual baselines for AfriSenti on the 12 languages with training splits. Results are averaged over 5 runs.

TaV3 (He et al., 2021). For the Africa-centric models, we made use of AfriBERTa-large (Ogueji et al., 2021a) and AfroXLMR-{base & large} (Alabi et al., 2022) — an XLM-R model adapted to African languages. AfriBERTa was pre-trained from scratch on 11 African languages including nine of the AfriSenti languages while AfroXLMR supports 10 of the AfriSenti languages. Additionally, we fine-tune XLM-T (Barbieri et al., 2022a), an XLM-R model adapted to the multilingual Twitter domain, supporting over 30 languages but fewer African languages due to a lack of coverage by Twitter's language API (cf. §4).

### 6.2 Experimental Results

Table 7 shows the results of the monolingual baseline models on AfriSenti. AfriBERTa obtained the worst performance on average (61.7), especially for languages it was not pre-trained on (e.g., < 50 for the Arabic dialects) in contrast to the languages it was pre-trained on, such as hau, ibo, swa, yor. XLM-R-base led to a performance comparable to AfriBERTa on average, performed worse for most African languages except for the Arabic dialects and pt-MZ. On the other hand, AfroXLMR-base and mDeBERTaV3 achieve similar performances, although AfroXLMR-base performs slightly better for kin and pcm compared to other models.

Overall, considering models with up to 270M parameters, XLM-T achieves the best performance which highlights the importance of domain-specific pre-training. XLM-T performs particularly well on Arabic and Portuguese dialects, i.e., arq, ary and pt-MZ, where it outperforms AfriBERTa by 21.8, 14.2, and 13.0 and AfroXLMR-base by 4.0, 5.9, and 4.7 F1 points respectively. AfroXLMR-large achieves the best overall performance and improves over XLM-T by 2.5 F1 points, which shows the

| Model | F1 |
|---|---|
| AfriBERTa-large | 64.7 |
| XLM-R-base | 64.3 |
| AfroXLMR-base | 68.4 |
| mDeBERTaV3-base | 66.1 |
| XLM-T-base | 65.9 |
| XLM-R-large | 66.9 |
| AfroXLMR-large | **71.2** |

Table 8: Multilingual training and evaluation on combined test sets of all languages. We show the average scores over 5 runs.

benefit of scaling for large PLMs. Nevertheless, scaling is of limited use for XLM-R-large as it has not been pre-trained on many African languages.

Our results show the importance of both language and domain-specific pre-training and highlight the benefits of scale for appropriately pre-trained models.

Table 8 shows the performance of multilingual models fine-tuned on the combined training data and evaluated on the combined test data of all languages. Similarly to earlier, AfroXLMR-large achieves the best performance, outperforming AfroXLMR-base, XLM-R-large, and XLM-T-base by more than 2.5 F1 points.

Finally, Table 9 shows the zero-shot cross-lingual transfer performance from models trained on different source languages with available training data in the test-only languages orm and tir. The best source languages are Hausa or Amharic for orm and Hausa or Yorùbá for tir.

Interestingly, Hausa even outperforms a multilingually trained model. The impressive performance for transfer between Hausa and Oromo may be due to the fact that both are from the same language family and share a similar Latin script. Furthermore, Hausa has the largest training dataset in

| Source Language | Target Language | | |
| --- | --- | --- | --- |
| | orm | tir | AVG |
| amh | 46.5 | 62.6 | 54.6 |
| arq | 27.5 | 56.0 | 41.8 |
| ary | 42.5 | 58.6 | 50.6 |
| hau | **47.1** | **68.6** | **57.9** |
| ibo | 41.7 | 39.8 | 40.8 |
| kin | 43.6 | 64.8 | 54.2 |
| pcm | 26.7 | 58.2 | 42.5 |
| por | 28.7 | 21.5 | 25.1 |
| swa | 36.8 | 26.7 | 31.8 |
| tso | 21.5 | 15.8 | 18.7 |
| twi | 9.8 | 15.6 | 12.7 |
| yor | 39.2 | 67.1 | 53.2 |
| multilingual | 42.0 | 66.4 | 54.2 |

Table 9: Zero-shot evaluation on `orm` and `tir`. All source languages are trained on AfroXLMR-large.

AfriSenti. Both linguistic similarity and size of source language data have been shown to correlate with successful cross-lingual transfer (Lin et al., 2019).

However, it is unclear why Yorùbá performs particularly well for `tir` despite the difference in the script. One hypothesis is that Yorùbá may be a good source language in general, as claimed in Adelani et al. (2022) where Yorùbá was the second best source language for named entity recognition in African languages.

## 7    Conclusion and Future Work

We presented AfriSenti, a collection of sentiment Twitter datasets annotated by native speakers in 14 African languages: Amharic, Algerian Arabic, Hausa, Igbo, Kinyarwanda, Moroccan Arabic, Mozambican Portuguese, Nigerian Pidgin, Oromo, Swahili, Tigrinya, Twi, Xitsonga, and Yorùbá, used in the first Afro-centric SemEval shared task—SemEval 2023 Task 12: Sentiment analysis for African languages (AfriSenti-SemEval). We reported the challenges faced during data collection and annotation, in addition to experimental results in different settings.

We publicly release the datasets and other resources, such as the collection lexicons for the research community intetrested in sentiment analysis and under-represented languages. In the future, we plan to extend *AfriSenti* to additional African languages and other sentiment analysis sub-tasks.

## 8    Ethics Statement

Automatic sentiment analysis can be abused by those with the power to suppress dissent. Thus, we explicitly forbid the use of the datasets for commercial purposes or by state actors, unless explicitly approved by the dataset creators. Automatic sentiment systems are also not reliable at individual instance-level and are impacted by domain shifts. Therefore, systems trained on our datasets should not be used to make high-stakes decisions for individuals, such as in health applications. See Mohammad (2022, 2023) for a comprehensive discussion of ethical considerations relevant to sentiment and emotion analysis.

## 9    Limitations

When collecting the data, we deleted offensive tweets and controlled for the most conflicting ones by adding an annotation round or removing some tweets as explained in Section 5. We acknowledge that sentiment analysis is a subjective task and, therefore, our data can still suffer from the label bias that most datasets suffer from. However, we share all the attributed labels to mitigate this problem and help the research community interested in studying the disagreements.

Given the scarcity of data in African languages, we had to rely on keywords and geographic locations for data collection. Hence, our datasets are sometimes imbalanced, as shown in Figure 3.

Finally, although we focused on 14 languages, we intend to collect data for more languages. We invite the community to extend our datasets and improve on them.

## Acknowledgements

We thank all the volunteer annotators involved in this project. Without their support and valuable contributions, this project would not have been possible. This research was partly funded by the Lacuna Fund, an initiative co-founded by The Rockefeller Foundation, Google.org, and Canada's International Development Research Centre.

The views expressed herein do not necessarily represent those of Lacuna Fund, its Steering Committee, its funders, or Meridian Institute. We are grateful to Adnan Ozturel for helpful comments on a draft of this paper. We thank Tal Perry for providing the LightTag (Perry, 2021) annotation tool. We also thank the Language Technology Group, University of Hamburg, for allowing us to use the WebAnno (Yimam et al., 2013) annotation tool for all the Ethiopian languages annotation tasks. David Adelani acknowledges the support of DeepMind

Academic Fellowship Programme. Finally, we are grateful for the support of Masakhane.

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

# A  Appendix

| Lang. | Tweet | Sentiment |
|---|---|---|
| amh | ያ ጨካኝ አረመኔ ታስሮ ይሽዉ ካቴና ገብቶለታል ይሉናል። ቆይ አስረዉ የጀበና ቡና እያጋበዙት ነዉ እንዴ?
**Gloss:** They are telling us that the cruel barbarian is behind the bar and got chains. Wait! Are they chaining him and servicing him coffee? | negative |
| arq | @user .... الشروق هذه من خرجت وهي تتاع تهديل، مستوى منحط وشعبوي
**Gloss:** Since it was founded, Echourouk [newspaper/TV channel] has always been shameful, low level and populist. | negative |
| ary | واش بغيتوهم يبداو يتكرفسو على العادي والبادي عاد تبقاو أنتا على خاطر خاطرڭم
**Gloss:** Do you want them to start being harsh on everyone to be relieved | negative |
| ary | rabi ykhali alhbiba makayn ghir nachat o chi machat
**Gloss:** God bless you, my dear let the fun begin | positive |
| hau | @USER Aunt rahma i luv u wallah irin totally dinnan
**Gloss:** @USER Aunty rahma I swear I love you very much. | positive |
| ibo | akowaro ya ofuma nne kai daalu nwanne mmadu
**Gloss:** they told it well my fellow sister well done at the end we will be all right | positive |
| kin | @user Ariko akokanu ngo inyebebe unyujijemo sisawa wangu
**Gloss:** @user but this thing of miscreant you just mentioned is not good dear | negative |
| orm | @user Jawaar Kenya OMN haala akkamiin argachuu dandeenya
**Gloss:** @USER Our Jewar how can we access/reach out OMN. | neutral |
| por | Honestidade é algo que não se compra. Infelizmente a humanidade esqueceu disso por causa das suas ambições.
**Gloss:** Honesty is something you can't buy. Unfortunately, humanity has forgotten this because of its ambitions. | positive |
| pcm | E don tay wey I don dey crush on this fine woman …
**Gloss:** I have had a crush on the beautiful woman for a while … | positive |
| swa | Asante sana watu wa Sirari jimbo la Tarime vijijini Huu ni Upendo usio na Mashaka kwa Mbunge wenu John Heche
**Gloss:** Thank you very much people of Sirari, rural Tarime province This is Undoubted Love for your Member of Parliament John Heche | positive |
| tir | @user ከመኸረኩም እንተኾይነ፡ንሕ ውሓት ነዞም ውሑድ ቄጽሮም እባ ምጥፋእ ይሕሸ ኩም!
**Gloss:** If I were to advise you:you better get rid of these few | negative |
| tso | @user @user Yu , tindzava ? Tsika mbangui mpfana e nita ku desprogramara
**Gloss:** Ah! gossiping?  Quit drugs dude, they'll mess you up… | negative |
| twi | messi saf den check en bp na wo kwame danso wo di twe da kor aaa na wawu | negative |
| yor | onírèègbè aláàdúgbò ati olójúkòkòrò
**Gloss:** mischievous and coveteous neighbour | negative |

Table 10: Annotated tweets with their English translations.