# OpenReview forum: "AfriSenti: A Twitter Sentiment  Analysis Benchmark for African Languages"
_EMNLP/2023/Conference — EMNLP 2023 Main_

### Official Review · Reviewer_j4yL · 2023-07-31

**Soundness:** 3

**Excitement:**

3: Ambivalent: It has merits (e.g., it reports state-of-the-art results, the idea is nice), but there are key weaknesses (e.g., it describes incremental work), and it can significantly benefit from another round of revision. However, I won't object to accepting it if my co-reviewers champion it.

**Paper Topic And Main Contributions:**

This paper introduces a new Twitter benchmark for sentiment analysis in african languages, and provide experimental results with few baseline models.

**Reasons To Accept:**

- A new useful dataset that can be used for sentiment analysis in Afriacan languages.
- Extensive experimental results on the benchmark.

**Reasons To Reject:**

- Limited explanation of imbalanced data distribution in Table 6. Why there is no training samples in 'orm' and 'tir'? I see that there was a difficulty in collecting data from annotators, but it does not explain why we don't have training samples. For example, we can have 1500 training samples and 600 test samples in 'orm' language.
- More detailed experimental results are required. In Table 8, only F1 score is reported, but it is better to give per-label precision and recall as well.

**Reproducibility:**

3: Could reproduce the results with some difficulty. The settings of parameters are underspecified or subjectively determined; the training/evaluation data are not widely available.

**Reviewer Confidence:**

3: Pretty sure, but there's a chance I missed something. Although I have a good feel for this area in general, I did not carefully check the paper's details, e.g., the math, experimental design, or novelty.

---

> ### Author Rebuttal · Authors · 2023-08-28
>
> We would like to thank the reviewer for providing feedback. The main contribution of this work is to introduce new data resources for 14 African languages (from 3 language families), which is aligned with the EMNLP's main contribution: "new data resources, particularly for low-resource languages". We believe that our data resource is valuable for the NLP community, as it can enable new research directions and applications for these languages. Most importantly, it will encourage a new generation of African researchers who can work on languages that matter deeply to them and their communities.
>
> Below, we respond to the reviewer’s concerns..
>
> 1. With the additional page, if accepted, we will add a discussion of class imbalance for different languages, possible reasons for this, and their implications.
>
>
> 2. For orm and tig languages, we don't have enough training data. Therefore, we chose to conduct a zero-shot experiments for these languages.  In fact, Twitter is not the most commonly used platform in some African languages including Ethiopia, where most orm and tig speakers live. Hence, this was  an expected challenge in low-resource settings (less availability of suitable public data, difficulty in obtaining qualified annotators, etc.). Still, we are happy we could create larger datasets for 12 other languages, and at least some data for ‘orm’ and ‘tig’.
>
> 3. We will add the suggested per-label precision and recall in the revised version.

---

### Official Review · Reviewer_TvNi · 2023-08-03

**Soundness:** 4

**Excitement:**

4: Strong: This paper deepens the understanding of some phenomenon or lowers the barriers to an existing research direction.

**Paper Topic And Main Contributions:**

The authors present a dataset of 12 African languages, annotated for sentiment by native (and usually local) speakers. They describe challenges in collection and annotation. They also present some results from pre-trained language models, discussing which models do best with which languages and which source languages do the best in zero-shot learning for Oromo and Tigrinya.

**Questions For The Authors:**

I'm going to ask my questions in the order they pop up in the text, but I'll try to give a sense of which ones I think are more important to address:

1) You have picked sentiment and in lines 47-56, you mention a number of reasons that sentiment may be useful (literary analysis, culturonomics, commerce, psychology, social science). One that I don't think those quite capture would be "linguistic". Here, I'm thinking about how, for example, there's a malefactive particle in at least one Ethiopian language that lets a speaker/writer give the sense that someone did something AGAINST someone else, in Dutch there's "tet" which marks surprise/irritation, in Czech diphthongs can mean affection (or pejoration), in Cantonese, throwing a -k at the end of particles can intensify the emotion being expressed, in Tongan there are determiners that express sympathy, in Manambu there is a "frustrative" case marking for when things are done in vain, etc.

In my heart of hearts, my hope is that your dataset could help people find affective markers/word-order/morphemes but...is that plausible, do you think? Or is your expectation that what your annotators picked up on were louder words like adjectives ("awful", "wonderful", etc). I don't think this particularly needs to be added to your paper, but I am curious about how you'd look at the dataset being used for more linguistic/sociolinguistic/pragmatics research questions.

2) You mentioned using Saif Mohammad's sentiment guidelines, I think you mean the "base" questions he has (not the ones that try to get at what the sentiment is direct towards). It probably would be good to share the explicit instructions for annotators, probably in an appendix. This I do think is probably a minor weakness and worth addressing, btw. It also connects to some of your comments on code-switching texts—you have some annotators who don't know parts of those mixed language tweets and it feels like they should have a "I don't understand this" option (which would also be relevant for everyone else since I am certain as a native English speaker you could give me completely English tweets that I could not understand to give a sentiment to).

2b) Part of this question comes from looking at your examples in Table 1. The Amharic example kicks off with the very powerful bigram "ጨካኝ አረመኔ" (~brutal barbarian, pitiless pagan) but the rest of it has a much lighter tone as far as I can tell. I don't know the referent but it strikes me that this is fairly sarcastic and has an intent to be humorous which is "negative" in a way that is less clear than "I hate X". You mention sarcasm a number of times—as does Mohammad 2016—I'm wondering what you might say about that topic more generally.

This also feels related to the annotation problem you describe in Twi with “Tweaa” being potentially offensive depending on context—many annotators have a hard time distinguishing a speaker/writer’s sentiment when they use taboo terms (the terms that one community of speakers might think is unexceptional or just expressive may be no-go’s for other speakers who ignore the speaker intention). Do you have any thoughts about your annotators’ predilections on these things?

If I gave you a magic button to press that would replace positive/negative/neutral with any categories of your choosing with high accuracy, which categories would you pick? That is, is valence the real thing you most want or is it a stand-in/approximation/first-step for something you actually would rather collect but is harder?

3) You briefly mention some of the ways African languages present challenges for sentiment analysis that English and other languages may not. You mention tone, code-switching, and digraphia in lines 127-128 but I don't think you actually talk about tone. Here, I assume you mean phonemic tone since lots of African languages are tonal in nature. You discuss code-switching and digraphia but do you have anything to say about phonemic tone and your dataset/annotations? About half your languages use tone but I think only Yorùbá marks them in writing...but maybe the lack of marking them in tweets causes some extra homographs but...I'm not sure how that would be different than other languages which are replete with homographs (English lead=metal, lead=guide).

4) You have to do a bunch of things to get a meaningful (not-all-neutral) set of tweets. I think you do a pretty good job but I wonder if you could address what you see as the consequences of these choices for possible uses of the dataset. This is a suggestion I think would help the paper.

4b) A minor thing but why are the Nigerian language test sets lower-cased (you reference Muhammad et al 2022, but it’d be nice if you give the reason so readers don’t have to track that reference down).

4c) You discard “full disagreement” tweets—are those going to be in the dataset? I think that may help with the use of your dataset, provided it’s easy for people to see that these are more for agreement/annotator kinds of questions. I think your mention of Prabhakaran et al (2021) means that you probably do release these but it’s not completely clear to me.

4d) What do you think are the consequences of removing offensive tweets in Algerian Arabic? And was your intention more to produce a non-offensive dataset or to not disturb your annotators? I think it is relevant to share your intentions for this.

5) The idea that Yorùbá is a generally good source language strikes me as very strange. I mean, you are empirically showing it to be the case and mention Adelani et al’s similar finding in NER but...this strikes me as “surely a coincidence” or perhaps because there are some proper names with similar sentiment in Ethiopia and Nigeria…what do you think the most reasonable explanation of this is?

6) Your plan is to keep releasing more languages (and enable others to follow your lead), which is great. I wonder which languages are at the top of your list. Do you want coverage in terms of speaker populations? Coverage over countries that yet covered? Languages that are outside the language families you’ve already covered? I’m interested in your actual “Top Five” but even moreso for what leads you to that Top Five instead of another Top Five. I think being explicit about this helps readers understand where you're headed/why and what they should keep an eye out for or help with.


**Reasons To Accept:**

A very good reason to accept this paper is the dataset itself and the ability to talk about collection and annotation outside the parts of the world NLP researchers typically focus on. The authors share specific problems and their solutions that should also be good for discussion and help researchers who are interested in working on other lower resource languages.

They are also releasing the annotations, not just the majority-vote label, which should be useful for researchers interested in methodologies and inter-annotator agreement topics.

**Reasons To Reject:**

There are no risks for having this paper presented at the conference. In the next section, you'll see me ask a bunch of questions of the authors but I don't really consider these weaknesses, certainly not enough to jump up to a reason to reject.

**Reproducibility:**

3: Could reproduce the results with some difficulty. The settings of parameters are underspecified or subjectively determined; the training/evaluation data are not widely available.

**Reviewer Confidence:**

4: Quite sure. I tried to check the important points carefully. It's unlikely, though conceivable, that I missed something that should affect my ratings.

---

> ### Author Rebuttal · Authors · 2023-08-28
>
> We would like to thank the reviewer for providing feedback, and insightful suggestions. Below, we respond to the questions asked.
>
> 1.
>
> We appreciate the reviewer's insights on how the datasets can be useful to better understand how language is used in the African context and hope that researchers with expertise in linguistics find our dataset to be valuable.
>
> 2.
>
> (a): Yes, we are referring to the base questions. The one called, “A Simple Sentiment Questionnaire”. We will add this specificity in the paper. We will also add the annotation guide in the  appendix section. In some languages (e.g., Algerian Arabic), we added language-specific examples and slightly edited the guidelines, we will make these guidelines public as well.
>
> (b): Sarcasm: Since sarcasm was not a direct focus of this work, we have not explored it deeply yet. However, that ist interesting future work and we mentioned in lines 378-379 that sarcasm is generally one of the main challenges in our annotation.
>
> 3.
>
> Igbo is also a tonal language and marks them in writing. The tone in this language changes the sentiment. For example, below are identical tweets, but when used with different tones, they convey different sentiments.
>
> ò nwèkwàrà mgbe i naenwe sense ? ( Will you ever be able to talk sensibly? – You’re a fool.)
>
> ò nwèkwàrà mgbe i naenwe sense ( Sometimes you act with great maturity – I’m impressed.)
>
> We will add this in the revised paper.
>
> 4.
>
> (a)  We acknowledge that our dataset is not representative of the general distribution of sentiments in each language, as we applied various filtering and sampling methods to obtain a balanced and diverse set of tweets (we will add this point in the camera-ready). Therefore, someone looking at the dataset should not assume that random tweets for that language have the same distribution of pos/neg/neu as in the dataset for that language. Moreover, the performance of models trained on our dataset may vary depending on the target data they are applied to, especially if the target data has a very different distribution of classes.
>
> (b) The test set is already lower-cased in Muhammad et al. 2022. We will update this information in the paper.
>
> (c) The full disagreement data points are not part of our final dataset. However, we will release the individual annotator's label separately, including the full disagreement. This can be helpful for researchers interested in learning from disagreement.
>
> (d)  For offensive language instance, we did this for both reasons as offensive language detection was not the focus of this work and as in many languages these are the only labeled datasets (and thus may be used for many different tasks), we believe it is better to not include large amounts of offensive instances.
>
> 5. We intend to run more experiments in order to answer this question.
>
> 6. We do not have a top 5 but we intend to collaborate with speakers of other African languages and release more datasets.

---

### Official Review · Reviewer_DhE5 · 2023-08-06

**Soundness:** 4

**Excitement:**

4: Strong: This paper deepens the understanding of some phenomenon or lowers the barriers to an existing research direction.

**Paper Topic And Main Contributions:**

This paper describes a sentiment dataset creation for 14 African languages and performed several experiments to classify the tweets into sentiment categories.

**Questions For The Authors:**

1. Sentiment categories are not explicitly mentioned in the text. I can guess from Table 1 that there are three categories (positive, negative and neutral). The authors should provide the categories in the text.

2. More details on the models will be helpful. How the fine tunning was performed? What are the tunning parameters? The architecture of the system will be helpful.

3. As the F1-score for the system are no great, what can be done to improve the system performances?

4. What was the reason to test the cross-lingual system only on orm and tir languages?

5. Is there any statistics avaialble on percentage of code-switching and code-mixing? It will be great to have these information in the paper.

6. Are these trained models will also be avialble for the researchers?

**Reasons To Accept:**

1. This dataset with 14 African languages benefits the NLP community to develop sentiment classification tools.
2. The data collection challenges were described in detail which is useful for NLP researchers working for low-resource languages.

**Reasons To Reject:**

1. There is no innovation in implemetation for NLP development.

**Reproducibility:**

2: Would be hard pressed to reproduce the results. The contribution depends on data that are simply not available outside the author's institution or consortium; not enough details are provided.

**Reviewer Confidence:**

4: Quite sure. I tried to check the important points carefully. It's unlikely, though conceivable, that I missed something that should affect my ratings.

---

> ### Author Rebuttal · Authors · 2023-08-28
>
> We appreciate the reviewer's feedback, and we would like to address some concerns about the lack of innovation in our implementation.
>
> We agree that our implementation is not novel, but that was not our goal. The main contribution of this work is to introduce new data resources for 14 African languages (from 3 language families), which is aligned with the EMNLP's main contribution: "new data resources, particularly for low-resource languages" (see https://2023.emnlp.org/calls/main_conference_papers/#contributions)).
>
> We believe that our data resource is valuable for the NLP community, as it can enable new research directions and applications for these languages. Most importantly, it will encourage a new generation of African researchers who can work on languages that matter deeply to them and their communities. Responses to questions are below:
>
> 1. The sentiment categories used are indeed: positive, negative, and neutral. It was an oversight and we will explicitly mention these categories at multiple appropriate places in the text.
>
> 2. We will add the batch size, sequence length and other details in the camera ready. We used the standard configuration for text classification fine-tuning on HuggingFace sample codes, the only change is lowering the learning rate for the larger pre-trained language models  (1e-5).
>
> 3. Expanding our dataset with more data could improve. We believe that trying different techniques such as transfer learning and continuous training could also help increase the F1 scores.
>
> 4. We chose zero-shot setting for orm and tir because we were only able to collect a small amount of data. We decided to use the collected data for development and test split.
>
> 5. We will add statistics about the code-switching.
>
> 6. All models will be made public and shared on HuggingFace.

---

### Meta-Review · Area_Chair_kAVC · 2023-09-08

**Recommendation:** 5

**Metareview:**

This paper introduces 14 sentiment analysis datasets, comprised of Tweets from 14 African languages. It is thorough and detailed as indicated by its Soundness scores (two 4s and one 3), describing not only the data collection and annotation processes and experimental benchmarks but also the specific challenges associated with curating each dataset. Reviewers acknowledge the positive and far-reaching impact these discussions can have on future research in low-resource NLP, as indicated by the Excitement scores (two 4s, one 3). In addition to its contributions to the field of sentiment analysis, this paper may be of interest to those working on inter-annotator agreement topics, as full annotations are released alongside the majority vote label (Reviewer TvNi). Likewise, discussion of language-specific challenges and probable solutions may be of consequence to those interested in data collection methodologies. Minor individual concerns were raised by each reviewer (e.g. missing details regarding fine tuning, precision and recall, reasons for removing offensive Tweets), but all were adequately resolved during the author rebuttal period.

In light of this, only minor revisions, addressing reviewers’ comments and questions, need to be made to ensure this paper is camera ready.

---

### Decision · Program_Chairs · 2023-10-07

**Decision:**

Accept-Main

**Comment:**

This paper introduces 14 sentiment analysis datasets, comprised of Tweets from 14 African languages. It is thorough and detailed as indicated by its Soundness scores (two 4s and one 3), describing not only the data collection and annotation processes and experimental benchmarks but also the specific challenges associated with curating each dataset. Reviewers acknowledge the positive and far-reaching impact these discussions can have on future research in low-resource NLP, as indicated by the Excitement scores (two 4s, one 3). In addition to its contributions to the field of sentiment analysis, this paper may be of interest to those working on inter-annotator agreement topics, as full annotations are released alongside the majority vote label (Reviewer TvNi). Likewise, discussion of language-specific challenges and probable solutions may be of consequence to those interested in data collection methodologies. Minor individual concerns were raised by each reviewer (e.g. missing details regarding fine tuning, precision and recall, reasons for removing offensive Tweets), but all were adequately resolved during the author rebuttal period.

In light of this, only minor revisions, addressing reviewers’ comments and questions, need to be made to ensure this paper is camera ready.